# Study on Dynamic and Static Performance of a Micro Digital Hydraulic Valve

**DOI:** 10.3390/mi13050741

**Published:** 2022-05-07

**Authors:** Meisheng Yang

**Affiliations:** 1School of Mechanical Engineering, Anhui University of Technology, Maanshan 243002, China; yms20190072@ahut.edu.cn; 2State Key Laboratory of Fluid Power and Mechatronic Systems, Zhejiang University, Hangzhou 310027, China; 3Engineering Technology Research Center of Hydraulic Vibration Technology, Anhui University of Technology, Maanshan 243002, China

**Keywords:** digital hydraulic valve, control frequency, static performance, dynamic performance, test system

## Abstract

Previous researchers mostly carried out simulation research and scattered experimental research on the static and dynamic characteristics of the digital valve, but did not form a systematic and in-depth study on the characteristics of the valve. Based on expounding the basic principles and functions of the valve and the test system, this paper carries out the test research under various variables for three kinds of static characteristics, including pressure differential-flow characteristics, signal-pressure characteristics, and signal-flow characteristics. The optimal control frequency of the valve is obtained from the comprehensive consideration of linear interval, linearity, and hysteresis. Three methods are systematically used to deeply study the dynamic characteristics, and the influencing factors of test results under various test conditions are analyzed. Through the research of this paper, it can provide relevant performance parameters for taking the digital valve as the system control element in the next step, and lay the foundation for the accurate control of the system.

## 1. Introduction

The traditional proportional valve system and servo valve system are continuously controlled by analog voltage. These systems have high control precision and good dynamic performance, but at the same time, there are also some deficiencies such as low reliability, long commissioning cycle, and low anti-pollution ability [1,2].

Digital valve is a hydraulic component that controls the opening and closing of the valve through digital switching value. Since the core of the digital valve system is control units composed of multiple digital valves, the static and dynamic performance of a digital valve determines the performance of the whole digital valve system. Therefore, many researchers have carried out a lot of experimental research on the performance of the digital valve, and obtained some research results. It mainly includes two aspects: one is the development of a digital valve test system, the other is the test research of one valve performance.

Puumala has developed a digital valve test system, which tests digital valves with different flow rates by arranging three flow meters with different ranges [3]. The pressure in front of the tested valve is set through the proportional overflow valve, the back pressure behind the tested valve is provided through the throttle valve, and the pressure fluctuation at the front and rear sides of the tested valve is maintained within a certain range through the accumulator. The dSPACE real-time system platform is used to build the control system. To isolate the influence of temperature on the test system, the system oil temperature is controlled at a constant condition of about 40 °C [4]. Paloniitty studied the micro water hydraulic on-off valve and developed a test system similar to the oil hydraulic digital on-off valve. In the system, the orifice is used to adjust the back pressure of the tested valve. Fluke i30 AC/DC clamp meter (Fluke company, Everide, Washington, DC, USA) is used for current detection, LV25-P Hall voltage sensor of LEM company(Beijing, China) is used to detect pulse voltage. The hardware of the measurement and control system adopts dSPACE DS1102 board to collect and control the test data. In the test study, the dynamic performance of current, voltage and valve outlet side pressure were tested [5]. Uusitalo developed a new type of bistable high-speed digital valve in 2010 and conducted experimental research. Due to the small flow of single valve, a group of valves were tested as a tested valve [6]. The front and rear sides of the tested valve are regulated by throttle valves, and the pressure fluctuation on both sides of the accumulator is maintained within a certain range.

Karvonen studied the miniaturization of digital valve and developed the first generation micro valve with an outer diameter of 10 mm. In the research, the flow field, electromagnetic field, and excitation circuit were optimized, and reasonable matching parameters were obtained [7]. Due to cavitation under a large differential pressure test, the pressure setting is lower than 12.5 MPa, the opening time is 1.9 ms and the closing time is 2.2 ms. Due to using stainless steel material with poor magnetic isolation performance in the magnetic circuit, there is an “open circuit” in the magnetic circuit. To avoid reducing the magnetic permeability, the electrical pure iron was not hardened at the end of the valve core, and the hardness of this material is low, so the durability of the valve core is poor. Karvonen studied the durability of the valve core and hardened the end of the valve core [8]. Paloniitty developed the third generation micro solenoid valve, and the outer diameter of this micro valve was redesigned to 10 mm. The assembly process of the valve has been greatly improved, and the coil is supported by a stainless steel framework. Under the test pressure of 21 MPa, the opening response time is 2.0 ms and the closing response time is 2.8 ms. Individual differences in valve closing delay are large [9]. Linjama developed the fourth-generation micro solenoid valve with an outer diameter of 10 mm. Compared with the stainless steel coil framework used in the third generation micro valve, the fourth generation valve coil is made of polyacetal resin to prevent the “open circuit” of the magnetic circuit. Cobalt iron alloy is used as soft magnetic material. Under different pressure drop, the opening delay time is 1.4~2.3 ms and the closing delay time is 2.0~3.4 ms. The limitation of the fourth generation micro valve is still the increase of working pressure. The current working pressure is 21 MPa [10]. Werely has tested the unidirectional performance of GMM actuators with different lengths (51 mm and 102 mm). When the excitation frequency was 325 Hz, they can achieve the maximum no-load output speed (84 mm/s and 88 mm/s) [11,12]. Oak Ridge National Key Laboratory proposed a digital hydraulic valve using memory alloy material. This valve used voltage to excite the memory alloy material to deform and expanded in the length direction, to control the opening and closing of the hydraulic valve. The pressure reaches 20.7 MPa, and the dynamic response of the valve can reach 200 Hz. It can control human-like manipulators and is also suitable for special medical devices, Since the flow of the memory alloy valve is only 10 mL/min, this microflow valve is difficult to meet the flow demand of the traditional hydraulic systems [13].

Caterpillar has developed a cone valve with a hollow valve core, which improved the overall response speed by reducing the motion quality of the valve core, and the opening and closing time is about 1ms. However, the front and rear seats of the valve core have high requirements for coaxial, difficult processing, and high cost of the valve. A normally open two position three-way solenoid valve for high-pressure common rail electronically controlled fuel injection system is developed by Nippon Denso. The valve sleeve and valve core of this valve were outer valve and inner valve, respectively. The inner valve was a plunger structure, and the outer valve and armature were fixed together. The opening and closing time were less than 1 ms [14]. Used numerical simulation and experimental research methods, Wang analyzed the influence of iron core cross-sectional area and electromagnetic coil parameters on electromagnetic force, and made a detailed analysis on the opening and closing characteristics of parameters such as opening current, holding current, spring stiffness and spring preload, and obtained the influence of relevant design variables on the electromagnetic force and dynamic characteristics of high-speed valve [15,16,17,18].

Early researchers have carried out a lot of work on design and testing of the digital valve, but they have not made systematic research [19,20]. This research starts from the introduction of valve and test system, deeply studies the test of static and dynamic characteristics under various test conditions, and analyzes a variety of test factors.

## 2. Digital Valve and Test System

This research takes the micro high-speed on-off valve developed in the early stage as the object, and its main structural parameters and electromagnetic performance have been deeply studied [21,22]. The structure of the digital valve is shown in Figure 1, which adopts electromagnetic actuator. The valve spool and valve cone are of segmented structure. In order to increase the electromagnetic force, the spring is arranged at the top, and the valve spool adopts soft magnetic material DT4. In order to increase wear resistance between the valve port and the valve cone, the valve cone is made of alloy steel Cr12MoV. In order to ensure the pressure balance before and after the valve, the valve spool opens the oil groove. In the process of movement, the current passes through the wire, causing the magnetic material to produce an electromagnetic field. Under the action of electromagnetic force, the moving iron spool overcomes friction, spring force and hydraulic force, and moves back and forth. Thus, the valve port can be opened or closed. The actual object is shown in Figure 2, and its supporting electronic control board is shown in Figure 3. In the previous research, the influence of multiple parameters on the electromagnetic force, magnetic field strength, magnetic field strength and signal flow characteristics of single valve has been studied, which provides a theoretical and experimental basis for research of this paper.

The system of the digital valve was also developed, and the working principle of the relevant system was described [23]. The hydraulic scheme of the test system is shown in Figure 4, including main circuit system and auxiliary circuit system. Among them, the pressure of the main circuit system is set through the proportional overflow valve 5. The test is completed in the main circuit system. The auxiliary circuit is mainly to compensate the leakage of the main circuit system and ensure the stability of the pressure behind the tested valve. The flow of the tested valve is measured by a flowmeter 15. The pressure is measured by the pressure sensors 12 and 14. As shown in Figure 5, the system is divided into two parts: the main circuit system and the auxiliary circuit system. Among them, the main circuit system is the main test circuit, which realized the functions of setting the pressure of the valve, stabilizing overflow, flow measurement and so on. The main function of the auxiliary circuit system is to compensate the system leakage. This system had three characteristics: accuracy, under the condition of small valve flow, even if there is leakage in the system and internal leakage of the valve, the pressure behind the valve can still be stabilized and measured accurately due to the liquid replenishment of the auxiliary circuit system. Second, broadness, there are two branches in the circuit behind the valve, and the pressure behind the valve can be tested under two different test conditions: no-load and load; Third, it is generally used for other digital valve tests. The tested valve and its valve block are independent of the system valve block. The installation space can meet the digital valves of other sizes, and the system input signal is common.

The basic parameters of the system are shown in Table 1.

The control system scheme of the test bench is shown in Figure 6. The whole control system takes into account safety, reliability, economy and scalability. The operation status of the main components of the system can be reflected in the control system. The signals of sensors (temperature sensor, pressure sensor, control board current and flow sensor) are collected through the data acquisition card. At the same time, the analog signal is sent to the proportional relief valve to control the pressure of the valve, the PWM signal is sent to the control board to change the control signal of the digital valve, and the switching value signal is sent to the electromagnetic directional valve to switch the oil circuit. The system has an oil temperature alarm, system pressure alarm, oil filter pollution alarm, and other alarm functions. When the system fails or needs emergency shutdown, it can be cut off through the emergency stop button on the electric control cabinet to ensure the safety of the system and avoid damage to the system.

## 3. Static Performance Test of the Digital Valve

### 3.1. Pressure Differential-Flow Characteristics

The pressure differential-flow characteristics of the micro high-speed digital valve are tested in the study. The inlet pressure of the valve is kept stable through the closed-loop control of the proportional relief valve in front of the valve, and the outlet pressure changes with the closed-loop control of the proportional relief valve behind the valve. Control the inlet pressure in front of the valve at 210 bar and the outlet pressure behind the valve at 140~210 bar, so that the pressure differential at both ends of the valve changes between 0~70 bar (according to the test standard, set the total pressure of the valve to be reduced to one-third of the maximum oil supply pressure). Set the control frequency of the valve to 100 Hz, the duty cycle to 100%, the outlet pressure behind the valve increases from 140 bar to 210 bar, and then decreases from 210 bar to 140 bar. Test it by reciprocating twice. The characteristic curve is shown in Figure 7. From the test results, the pressure differential-flow characteristics is stable and has good repeatability.

### 3.2. Control Signal-Pressure Characteristics

When testing the control signal-pressure characteristics, control the inlet pressure of the valve at 15 MPa, set the carrier frequency at 50 Hz, 100 Hz, 200 Hz and 400 Hz, respectively, and test the pressure characteristics of the valve by changing the duty cycle. The characteristic curve is shown in Figure 8. At low frequency (50 Hz), the hysteresis of pressure characteristics curve is small (4.9%), and with the increase of carrier frequency (400 Hz), the hysteresis of pressure characteristic curve increases (18.6%). This is because under the low-frequency test conditions, the electromagnetic performance of the valve core can be better restored with the increase or decrease of the signal, while under the high-frequency test conditions, the electromagnetic performance of the valve core cannot be completely restored with the change of the signal, resulting in the change of the electromagnetic force and finally the large hysteresis of the pressure characteristics. In addition, at low frequency, the linearity of pressure characteristics is better than that at high frequency. This is due to the different duty cycle, and the linear relationship between the position of valve core and duty cycle is large.

### 3.3. Signal-Flow Characteristics

When testing the signal-flow characteristics, the pressure of the micro high-speed digital valve is closed-loop controlled, and the duty cycle is adjusted under different signal frequencies to test the signal flow characteristics of the valve.

Visually, when the duty cycle is small, there is a dead zone in the flow, which is due to the short time of high level, the current does not reach a certain value, and the valve core fails to overcome the motion resistance and is in a static state. When the duty cycle is large, there is a saturation zone in the flow, because the high-level time is long, the current remains at a large value, the recovery force of the valve core is less than the electromagnetic suction, and the valve core is open. The signal-flow characteristic curve is shown in Figure 9. To display the signal-flow characteristics at different frequencies, the characteristic curves at five frequencies are divided into five diagrams. In the test, the pressure differential of the valve is controlled at 3.5 MPa through the proportional relief valve at both ends of the valve. At low frequencies (5 Hz, 10 Hz), the dead zone and saturation zone are small, but the flow fluctuation is large. At high frequency (100 Hz, 150 Hz), the dead zone and nonlinear zone are large and the linear zone is small. For the signal-flow characteristics at different frequencies, considering the factors such as system flow, pressure fluctuation, dead zone, saturation zone, and linear zone range, it is ideal to control the carrier frequency of the micro high-speed digital valve at 50 Hz.

The comparison of key parameters of signal-flow characteristics under different frequencies is shown in Table 2. By comparing the linear range, linearity and hysteresis loop at each frequency, it can be seen that at low frequency (5 Hz, 10 Hz), the linear range is smaller, the linearity is better than that in high frequency band (100 Hz, 150 Hz), and the hysteresis is larger than that in high frequency band. The linear range of high frequency band is small, and the hysteresis and linearity are relatively large. At 50 Hz frequency, the linear range is relatively large, the linearity and hysteresis are relatively small, and the overall level reaches the level of traditional proportional valve. Therefore, considering the linear interval, linearity and hysteresis, it is more reasonable to select 50 Hz for the valve.

## 4. Dynamic Performance Test of the Digital Valve

Dynamic performance of the digital valve is to study the delay characteristics of opening and closing under different excitation signals. It can be characterized by spool displacement, pressure fluctuation and coil current. Among them, the most direct and accurate method is to detect the displacement of the valve core. Since the micro high-speed digital valve has small volume, limited installation space, and the quality of the valve core itself is small, the installation of contact sensor is equivalent to adding additional motion quality to the valve core, which will produce large deviation to the test data, which is not conducive to the accurate characterization of its dynamic characteristics. If the laser displacement sensor is used for non-contact indirect displacement measurement, although the dynamic characteristic accuracy of the digital valve obtained from the test is high, the price of the high-precision laser sensor is high. The laser beam of the laser sensor needs to contact the end face of the valve core, and the valve core is assembled in the valve body, and the valve body is installed on the valve block. Therefore, the displacement measurement carried out through the laser sensor is difficult to achieve.

The dynamic characteristics of the micro high-speed digital valve are considered from three test methods: dry current dynamic response test method, valve outlet pressure test method under step signal and wet current dynamic characteristic test method.

### 4.1. Test Method for Dynamic Response Characteristics of Dry Current

The dry current dynamic response characteristic test method used in this paper is to test the system without oil flow. This method can avoid the interference of various factors such as oil pressure and hydrodynamic force. Due to the different dynamic characteristics obtained by different pulse width duty cycle, if the duty cycle is too small, the valve core does not acquire enough electromagnetic energy, cannot overcome the external resistance such as friction and spring preload, the valve core has no action response, and the dynamic current cannot appear the inflection point when the valve core reaches the end. If the duty cycle is too large, during the closing process of the valve core, due to the excessive electromagnetic energy, the combined force of spring restoring force and friction force is difficult to overcome the electromagnetic force, so that the valve core cannot return to the initial position, and the inflection point of the dynamic current when the valve core reaches the initial position cannot appear. Therefore, in order to enable the valve core to be fully opened and fully closed, the paper selects the test condition of 50% duty cycle. The test curve of dry current dynamic characteristics is shown in Figure 10. Section AB on the current curve represents the delay time period of valve core opening, with a duration of 1.41 ms, section BC represents the delay time period of valve core opening movement, with a duration of 0.24 ms, and section DE represents the delay time period of valve core closing, with a duration of 0.31 ms. The section EF represents the delay time period of valve core closing movement, with a duration of 0.33 ms. The total opening time of valve core is 1.65 ms and the closing time of valve core is 0.64 ms. Study the influence of different duty cycle signals on dynamic characteristics. Select 40 Hz signal frequency, 20%, 30% and 40% duty cycle. The test curves of dry current dynamic characteristics are shown in Figure 11. Section AB (A′B′, A″B″) on the current curve represents the delay time period of valve core opening, section BC (B′C′, B″C″) represents the delay time period of valve core opening movement, and section DE (D′E′, D″E″) represents the delay time period of valve core closing, section EF (E′F′, E″F″) represents the delay time period of valve core closing movement. The duration of each period is shown in Table 3. It can be seen that the influence of control signals with the same frequency and different duty cycle on each dynamic response is almost negligible.

### 4.2. Wet Current Dynamic Characteristic Test Method

The wet current dynamic characteristic test method is to dynamically monitor the current when there is oil flow in the system. During the test, the control signal is continuously input to monitor the current dynamic response characteristics. When the inlet pressure is 14 MPa, the control signal is 40 Hz, the duty cycle is 80%, and the current response is detected. The test results are shown in Figure 12, which shows the dynamic response curve of the current. Similar to the traditional method of characterizing the displacement of the valve core by the current inflection point, the AB section on the current curve represents the delay time period of the opening of the valve core, the BC section represents the delay time period of the opening movement of the valve core, the DE section represents the delay time period of the closing movement of the valve core, and the EF section represents the delay time period of the closing movement of the valve core. Under the current detection method, the delay time of valve core opening current is 1.22 ms, the rise time of valve core opening current is 0.42 ms, the total time of valve core opening is 1.64 ms, the delay time of valve core closing current is 0.25 ms, the decline time of valve core closing current is 0.36 ms, and the total time of valve core closing is 0.61 ms.

Under the test condition of inlet pressure of 14 MPa, adjust different working frequencies to study the influence of different frequencies on current response. In the test, the working frequencies are 20 Hz, 40 Hz and 100 Hz, and the duty cycle is 40%. The current response test results are shown in Figure 13. Under the current test method, the total opening time of valve core at different frequencies is about 1.64ms, and the total closing time of valve core is about 1.08 ms. The control signals of different frequencies have little effect on the current response time of opening and closing.

Under the test condition that the inlet pressure is 14 MPa, adjust the different duty cycle of the control signal, and study the influence of different duty cycle on the current response. In the test, the working frequency is 40 Hz, and the duty cycle is adjusted to 30%, 40% and 50%, respectively. The current response test results are shown in Figure 14. Under the current test method, the total opening time of the valve core at different frequencies is about 1.84 ms, and the total closing time of the valve core is about 0.69 ms. The control signals with different duty cycle have little effect on the current response time of valve opening and closing.

### 4.3. Valve Outlet Pressure Test under Step Signal

The third dynamic characteristic characterization method is the method of pressure dynamic response time. The test schematic diagram is shown in Figure 15. The miniature high-speed digital valve is equivalent to a B-type half bridge composed of a variable valve port and a fixed hydraulic resistance. When the system pressure remains constant through the proportional relief valve, that is, the inlet pressure of the valve remains stable.

When the compressibility of the oil in the control chamber is ignored, the pressure is positively correlated with the opening area of the digital valve. Therefore, the dynamic response time of digital valve can be estimated accurately by monitoring the dynamic change of cavity pressure.

In the pressure dynamic performance test, the control signal frequency of the micro high-speed digital valve is 50 Hz, the system pressure is adjusted to 10 MPa, when the duty cycle signal is 0, the valve is closed, and when the duty cycle is 100%, the valve is opened. Figure 16 is the corresponding curve of working voltage and digital valve outlet pressure when the valve port is opened. In the process of valve changing from closed state to open state, the working voltage jumps from 0 V to 12 V at 1 ms, the pressure response delay is about 0.7 ms, the pressure rise time is about 0.1 ms, and the whole pressure rise time is 0.8 ms.

Adjust the inlet pressure of the digital valve to 6.5 MPa, 7.5 MPa and 8.5 MPa, respectively, and test the dynamic response of the opening and closing pressure. The test results are shown in Figure 17 and Figure 18, respectively. Under the condition of valve opening, with the increase of inlet pressure, pressure differential increases, and the pressure rise time increases; When the valve is closed, pressure differential increases and pressure drop time decreases with increasement of inlet pressure.

## 5. Conclusions

The dynamic and static characteristics of the digital valve play a key role in the performance of the whole hydraulic system. Based on this consideration, this paper focuses on the dynamic and static characteristics of the micro high-speed digital valve, and focuses on the following work:(1)This paper expounds the digital valve and test system, focuses on the functional principle and main parameters of the test system, and describes the framework of its control system.(2)The static characteristics are tested, respectively, including pressure differential-flow characteristics, signal-flow characteristics and signal-pressure characteristics. Among them, the pressure differential-flow characteristic is good and the repeatability is high; The internal causes of signal-pressure characteristics in different sections are analyzed; From the comprehensive consideration of linear interval, linearity and hysteresis, we acquire a useful conclusion that it is reasonable to choose 50 Hz as the control frequency.(3)The principles of three dynamic characteristic test methods are described, namely, dry current dynamic response characteristic test, valve outlet pressure test under step signal and wet current dynamic characteristic test. The three experimental tests are carried out under different test conditions, and the influencing factors of the experimental results are analyzed.

## Figures and Tables

**Figure 1 micromachines-13-00741-f001:**
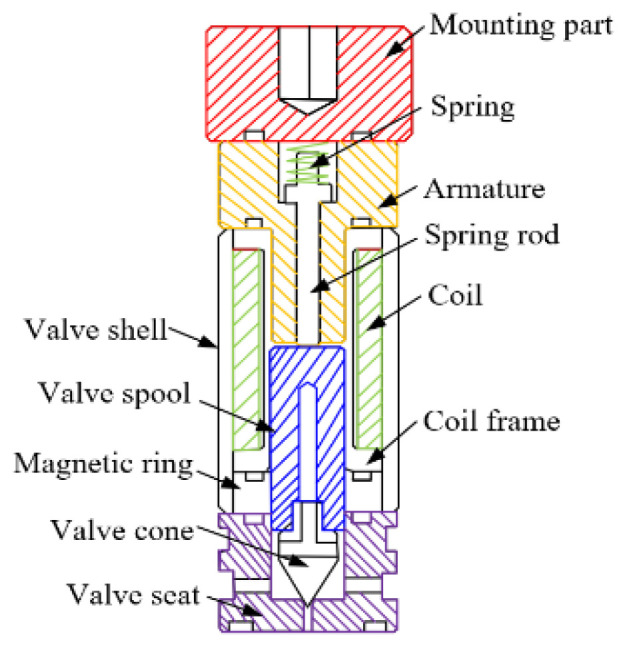
Structure of the micro high speed digital valve.

**Figure 2 micromachines-13-00741-f002:**
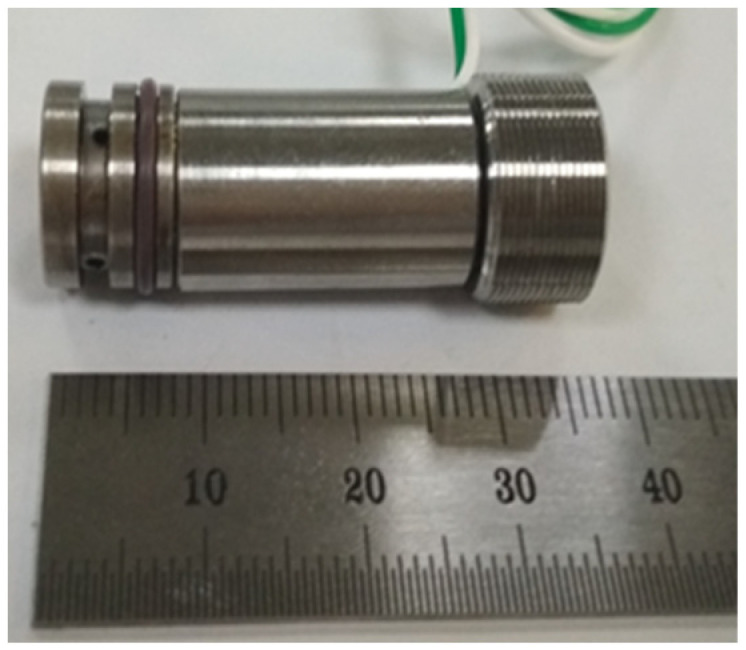
Prototype of the micro high speed digital valve.

**Figure 3 micromachines-13-00741-f003:**
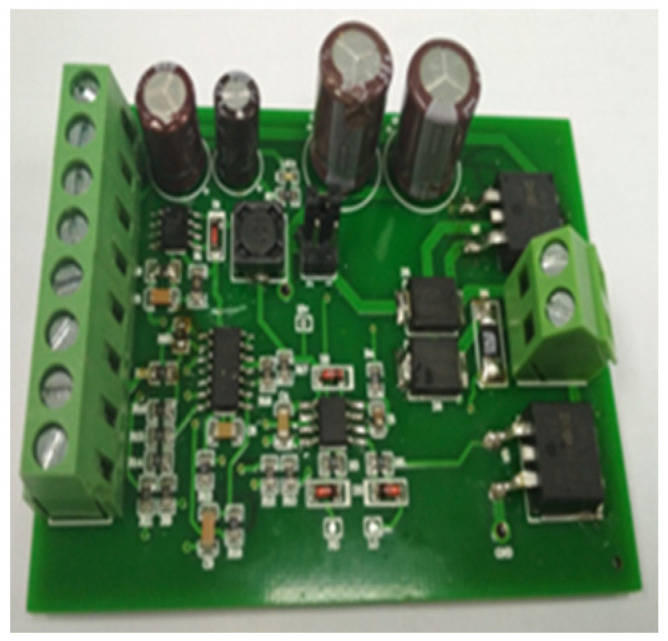
Digital valve electronic control board.

**Figure 4 micromachines-13-00741-f004:**
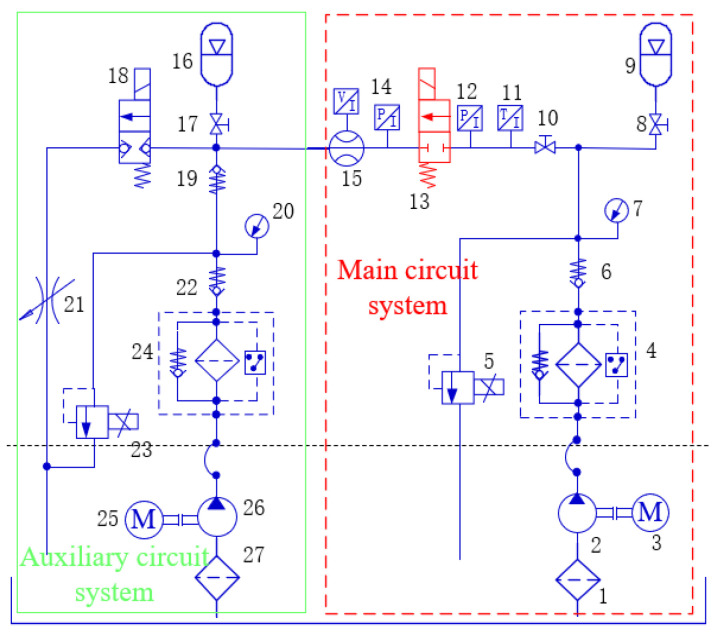
Hydraulic scheme of digital valve test system. 1. Oil suction filter. 2. Main oil pump. 3. Motor. 4. High pressure filter. 5. Proportional overflow valve. 6. Check valve. 7. Pressure gauge. 8. Stop valve. 9. Accumulator. 10. Stop valve. 11. Temperature sensor. 12. Pressure sensor. 13. Micro high speed digital valve. 14. Pressure sensor. 15. Flowmeter. 16. Accumulator. 17. Stop valve. 18. Solenoid on-off valve. 19. Check valve. 20. Pressure gauge. 21. Throttle valve. 22. Check valve. 23. Proportional overflow valve. 24. High pressure filter. 25. Motor. 26. Auxiliary pump. 27. Oil suction filter.

**Figure 5 micromachines-13-00741-f005:**
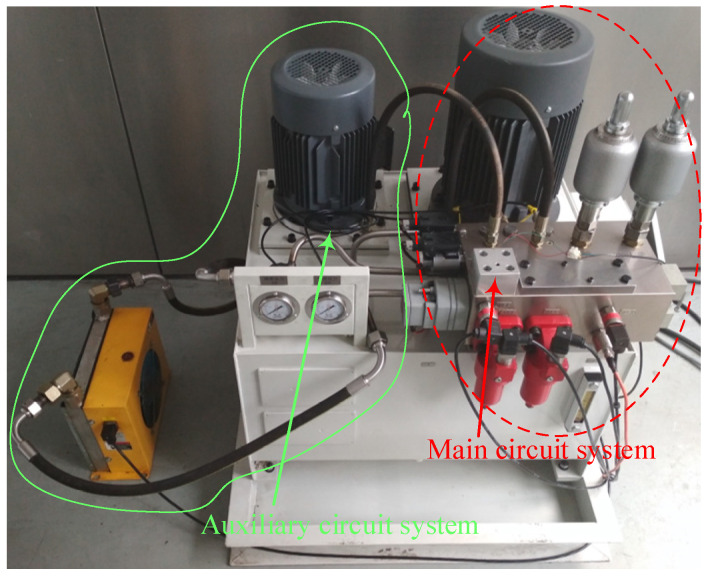
Digital valve test system.

**Figure 6 micromachines-13-00741-f006:**
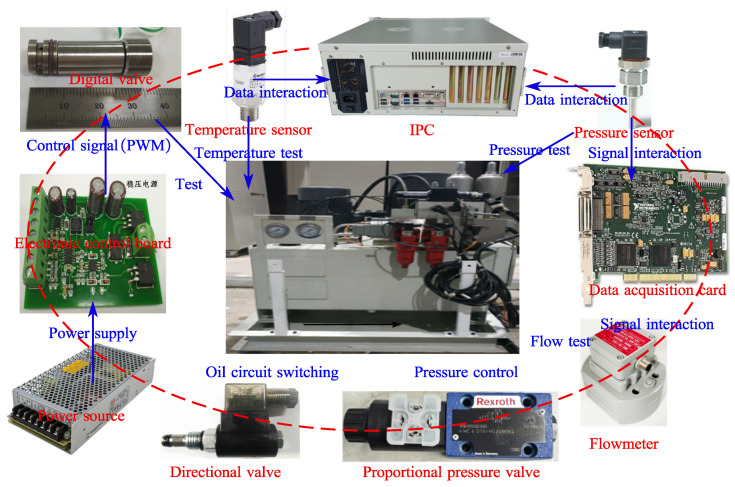
Measurement and control system scheme.

**Figure 7 micromachines-13-00741-f007:**
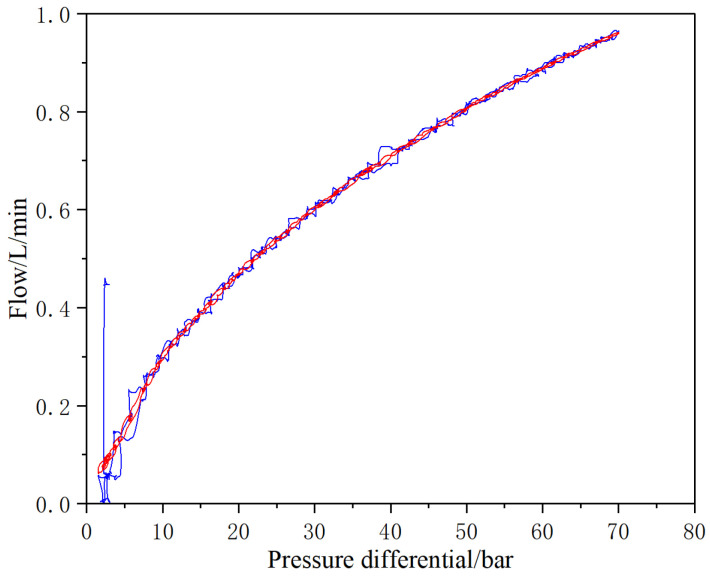
Pressure differential-flow characteristics curve.

**Figure 8 micromachines-13-00741-f008:**
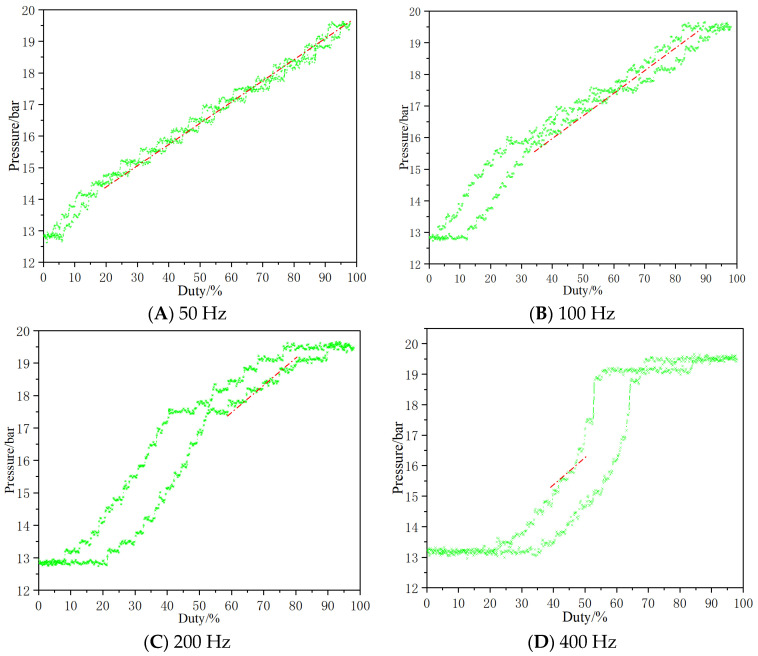
Signal-pressure characteristics curve.

**Figure 9 micromachines-13-00741-f009:**
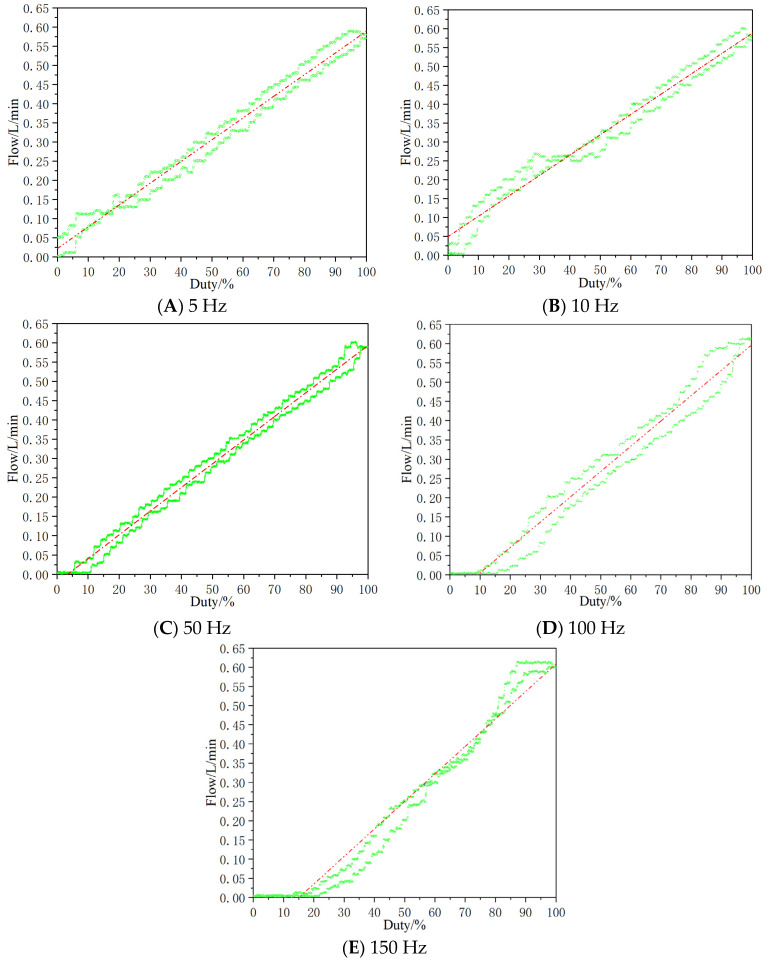
Signal-flow characteristics curve.

**Figure 10 micromachines-13-00741-f010:**
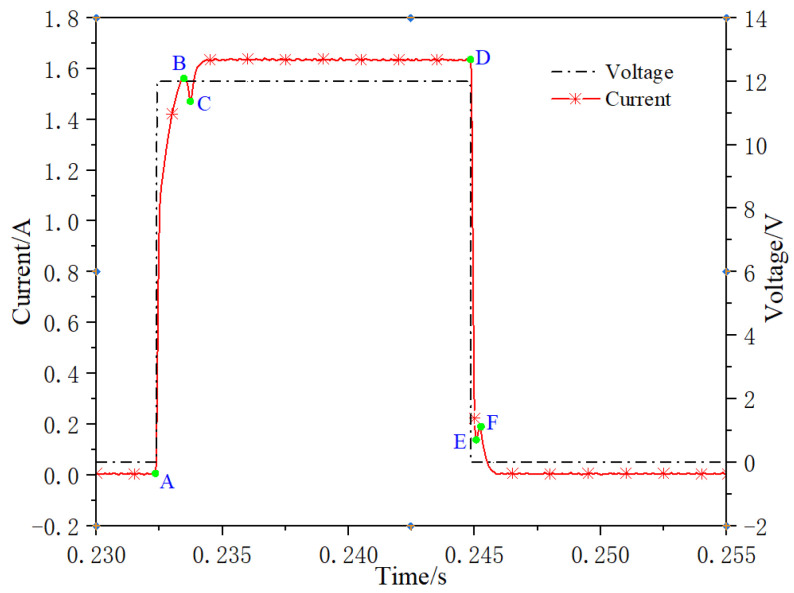
Dry current dynamic characteristics.

**Figure 11 micromachines-13-00741-f011:**
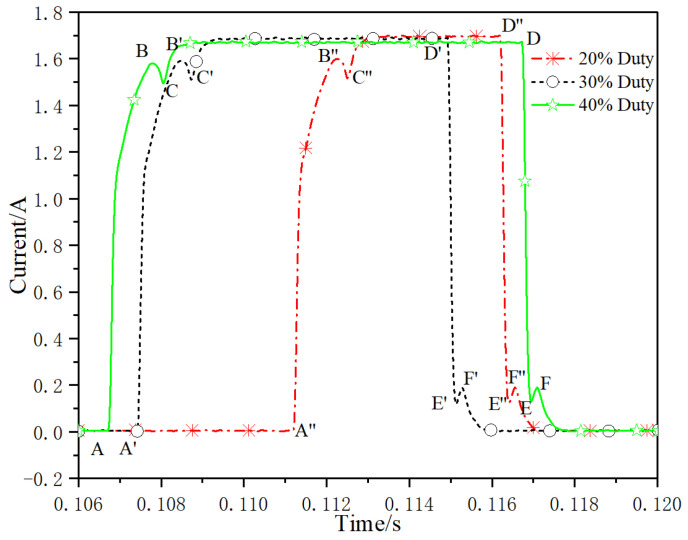
Dynamic response of dry current under different duty.

**Figure 12 micromachines-13-00741-f012:**
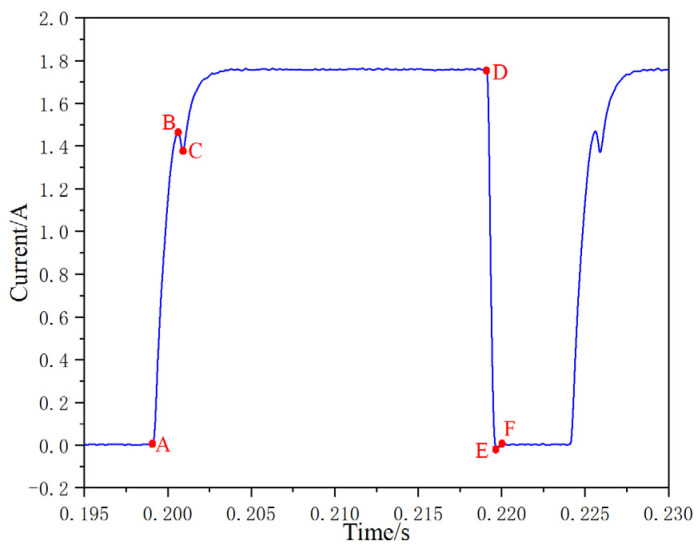
Dynamic response curve of wet current.

**Figure 13 micromachines-13-00741-f013:**
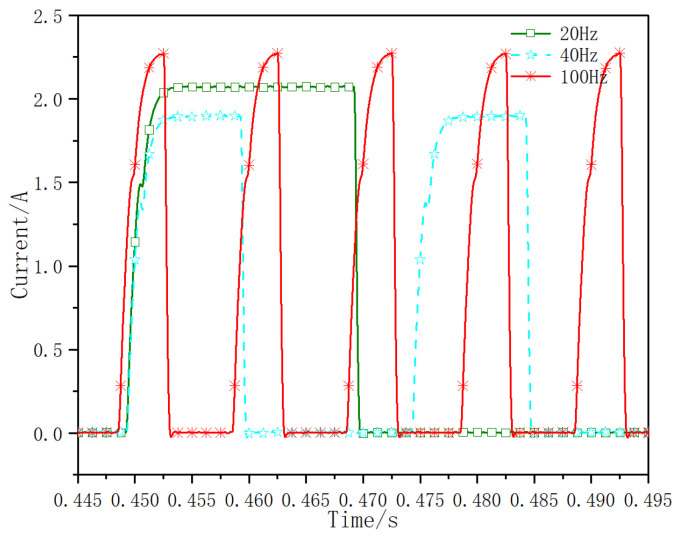
Current dynamic response curves at different frequencies.

**Figure 14 micromachines-13-00741-f014:**
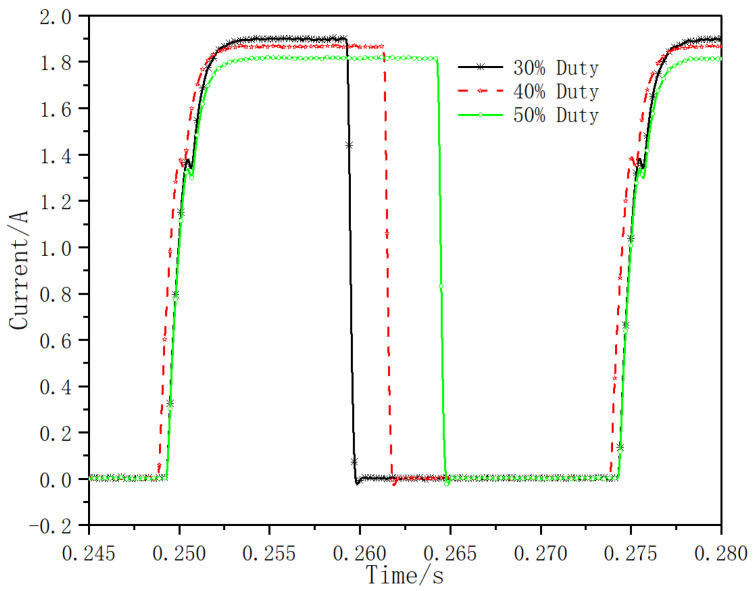
Current dynamic response curve under different duty cycle.

**Figure 15 micromachines-13-00741-f015:**
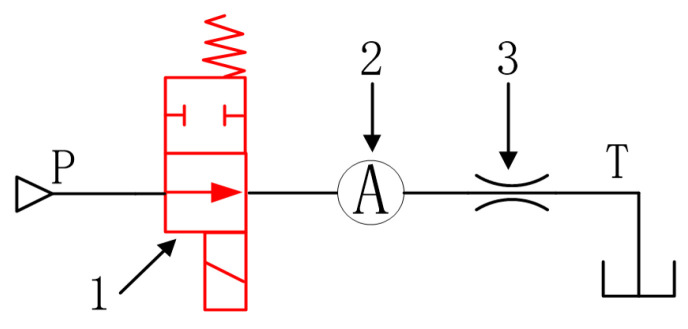
Schematic diagram of pressure dynamic performance test of Digital valve. 1. Digital valve. 2. Control chamber. 3. Fixed orifice.

**Figure 16 micromachines-13-00741-f016:**
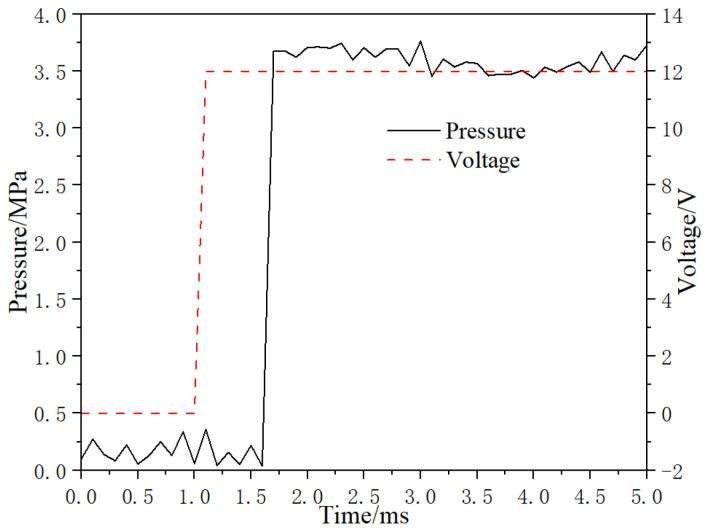
Dynamic response of opening pressure.

**Figure 17 micromachines-13-00741-f017:**
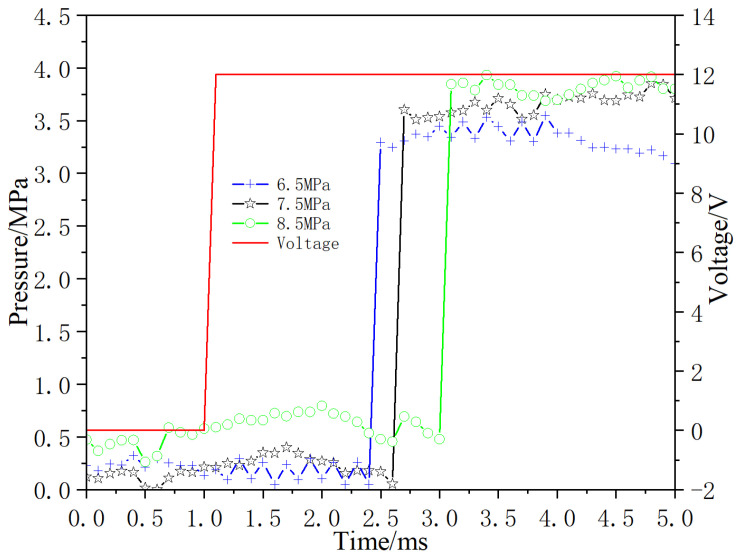
Dynamic response diagram of opening under different inlet pressure.

**Figure 18 micromachines-13-00741-f018:**
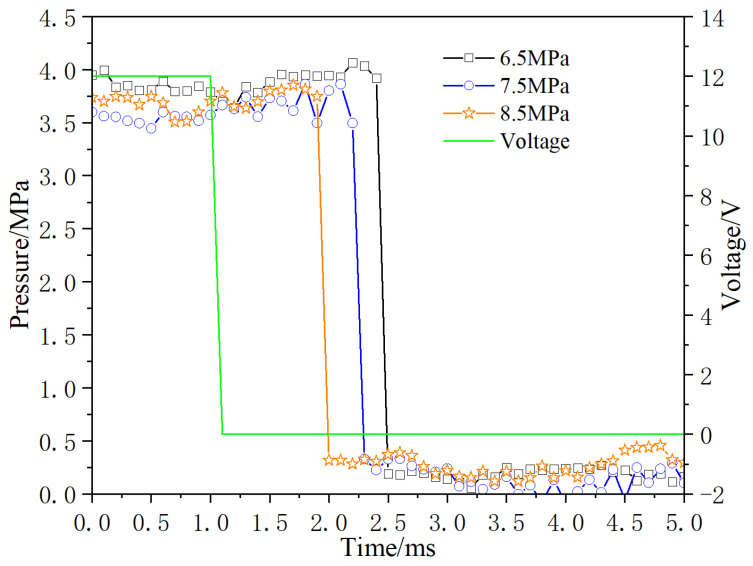
Dynamic response diagram of closing under different inlet pressure.

**Table 1 micromachines-13-00741-t001:** Basic parameters of digital valve test system.

Parameters	Value	Unit
Rated flow of the main pump	5.7	L/min
Rated pressure of the main pump	260	bar
Power of the main motor	4	kW
Rated flow of the auxiliary pump	3.1	L/min
Rated pressure of the auxiliary pump	260	bar
Power of the auxiliary motor	2.2	kW
Voltage of the strong current cabinet	380	V
Voltage of the control system	24/12	V

**Table 2 micromachines-13-00741-t002:** Key parameters of signal-flow characteristics under different frequencies.

	Frequency	5 Hz	10 Hz	50 Hz	100 Hz	150 Hz
Parameters						
Linear interval		30~98%	45~98%	10~95%	40~90%	60~75%
Linearity		8.6%	16.4%	3.2%	9.5%	21.3%
Hysteresis loop		12.3%	18.2%	6.8%	13.4%	7.8%

**Table 3 micromachines-13-00741-t003:** Dynamic response under different duty cycle.

	20% Duty				30% Duty				40% Duty			
Time section	AB	BC	DE	EF	A′B′	B′C′	D′E′	E′F′	A″B″	B″C″	D″E″	E″F″
Time/ms	1.38	0.28	0.35	0.26	1.36	0.29	0.36	0.27	1.4	0.23	0.33	0.3
Total time/ms	1.66		0.61		1.65		0.63		1.63		0.63	

## Data Availability

The data used to support the findings of this study are available from the corresponding author upon request.

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
