# Peer review of "Study on Dynamic and Static Performance of a Micro Digital Hydraulic Valve"

_micromachines, 2022, doi:10.3390/mi13050741_

Round 1
Reviewer 1 Report
It is an interesting paper on the actual topic. It is devoted to experimental research of static and dynamic characteristics of the micro digital hydraulic valve.
There are several comments that need to be addressed in the final version of the paper.
- English must be edited carefully.
- The introduction indicates the names of the authors of the reference papers, but does not provide links to their publications, e.g., Linjama (line 38). In some other cases, the links are too far from the first mentions of the works.
- Line 120: Ref [23] is not presented in the reference list.
- The valve design is not presented.
- In Line 120, it is stated “As shown in Fig. 3, the system is divided into two parts”. But these parts are not shown in the figure.
- Figure 3 shows only a photo of the test rig, but its individual units are not indicated, its structural and hydraulic schemes are not presented.
- There are too many arrows in Fig. 4. Their purpose is not clear. What do they indicate, the place where this element is installed or the direction of signal transmission?
- Measurement and control system circuits, the hydraulic system diagram should be presented to better understanding the testing procedure.
Author Response
Thank you very much for your valuable comments and suggestions which would be of great help to improve quality of our manuscript. I have modified the manuscript accordingly, and the detailed modifications in the revision are listed below point by point. The questions and comments of the Reviewers are marked in blue, and the responses from the authors are marked in black. The modifications in the revision are marked by a yellow highlight tool.
It is an interesting paper on the actual topic. It is devoted to experimental research of static and dynamic characteristics of the micro digital hydraulic valve.
There are several comments that need to be addressed in the final version of the paper.
*1: English must be edited carefully.
Responses *1: Thank you very much for your suggestion. Language of the paper has been revised again in the revised manuscript. Unreasonable English has been revised.
*2: The introduction indicates the names of the authors of the reference papers, but does not provide links to their publications, e.g., Linjama (line 38). In some other cases, the links are too far from the first mentions of the works.
Responses *2: The literature introduction on line 38 has been adjusted, and the publication links of the relevant references have been given in the revised manuscript.
*3: Line 120: Ref [23] is not presented in the reference list.
Responses *3: Ref [23] has been presented in the reference list of the revised manuscript.
*4: The valve design is not presented.
Responses *4: The structural design of the valve has been described in detail in the revised draft
*5: In Line 120, it is stated “As shown in Fig. 3, the system is divided into two parts”. But these parts are not shown in the figure.
Responses *5: In order to express these two parts clearly, the original figure has been replaced with a new one in the revised version.
*6: Figure 3 shows only a photo of the test rig, but its individual units are not indicated, its structural and hydraulic schemes are not presented.
Responses *6: Thank you very much for your suggestion. In the revised draft, the schematic diagram and relevant statements of the hydraulic scheme of the test system are added.
The detailed reply is as follows: The hydraulic scheme of the test system is shown in Fig. 4, including main circuit system and auxiliary circuit system. Among them, the pressure of the main circuit system is set through the proportional overflow valve 5. The test is completed in the main circuit system. The auxiliary circuit is mainly to compensate the leakage of the main circuit system and ensure the stability of the pressure behind the tested valve. The flow of the tested valve is measured by a flowmeter 15. The pressure is measured by the pressure sensors 12 and 14.
*7: There are too many arrows in Fig. 4. Their purpose is not clear. What do they indicate, the place where this element is installed or the direction of signal transmission?
Responses *7: Thank you very much for your suggestion. I have deleted some arrows in the revised version. Some arrows are related. For example, the temperature sensor tests the system temperature. The power source supplies power to the electronic control board. The electronic control board provides control signal to the tested valve. The pressure sensor and the temperature sensor transmit the data to the IPC.
*8: Measurement and control system circuits, the hydraulic system diagram should be presented to better understanding the testing procedure.
Responses *8: The hydraulic system scheme has been provided in the revised draft. The system studied in this paper is mainly realized by industrial computer and acquisition board through MATLAB and Labview programming language. The testing procedure are realized through the control program. The test equipment and control hardware are commercial products. Therefore, measurement and control system circuits have little significance for this study.
Thank you very much for your suggestion.

Reviewer 2 Report
Based on expounding the basic principles and functions of the valve and the test system, the autor in this paper carries out the test research under various variables for three kinds of static characteristics, including pressure differential-flow characteristics, signal-pressure characteristics, and signal-flow characteristics.
The article is chronologically arranged and has a logical sequence. The author describes in detail the research of other authors in the introduction.
In the second chapter, the author describes in detail the digital value and the test system, including the basic parameters of the digital valve of the test system. The Figure 4 describes well the various important components of the test system.
The results of the experimental solution are clearly processed in the third chapter. In the fourth chapter, the author deals with the dynamic performance test of the digital valve.
At the end of the article, the author clearly summarizes the whole article in three points.
Acknowledgments also testify to the fact that the topic of the article is current, where there are thanks to three projects, thanks to which this experimental solution was created.
I think that this article will be of benefit to many colleagues who deal with similar issues.
Author Response
Thank you very much for your valuable comments and suggestions.